# Patients and Communities Shape Regional Health Research Priorities: A Participatory Study from South Tyrol, Italy

**DOI:** 10.3390/healthcare13212797

**Published:** 2025-11-04

**Authors:** Christian J. Wiedermann, Verena Barbieri, Angelika Mahlknecht, Carla Felderer, Giuliano Piccoliori, Doris Hager von Strobele-Prainsack, Adolf Engl

**Affiliations:** Institute of General Practice and Public Health, Claudiana College of Health Professions, 39100 Bolzano, Italy

**Keywords:** health services research, research priority-setting, participatory research, patient and public involvement, primary care, South Tyrol, mental health, palliative care

## Abstract

**Highlights:**

**What are the main findings?**
Patient and social interest organizations identified research on the mental health of children and adolescents, continuity and trust in primary care, and patient-oriented palliative and end-of-life care as the top regional health services research priorities.General practitioners participated only marginally in regional health service research priority setting.

**What is the implication of the main findings?**
Regional health research agendas should incorporate community-driven priorities to ensure their relevance and uptake.Strategies are needed to improve general practitioner engagement in regional health services research to complement the active participation of patients and social organizations.

**Abstract:**

**Background/Objectives**: Engaging patients, caregivers, and community groups in health research priority-setting ensures that research agendas reflect genuine needs and enhance patient-centered care. Regions with cultural and linguistic diversity, such as South Tyrol in northern Italy, face challenges in achieving fair representation. This study aimed to identify health services research priorities in South Tyrol, a culturally and linguistically diverse region in Italy, through a bilingual participatory survey involving general practitioners (GPs) and patient and social interest organizations (PSIOs). **Methods**: A cross-sectional online survey (August–September 2025) was conducted among invited PSIOs (*n* = 64) and regional GPs (*n* = 290). A bilingual, self-developed questionnaire assessed organizational characteristics, priority ratings for predefined topics, experiences with research participation, and preferred participation modes. The data were analyzed descriptively. Group comparisons were performed using the Mann–Whitney U and chi-square tests with effect size calculation. Associations were examined using Spearman’s correlation. Free-text responses were thematically content-coded. **Results**: Ninety-five responses were analyzed, including nine general practitioners (9.5%) and 86 participants (90.5%) from patient and social interest organizations, of whom 27 (28.4%) held leadership or board positions. Across all groups, the highest-rated research priorities included children and adolescent mental health, palliative and end-of-life care, and continuity of primary care. Willingness to participate in future research was expressed by 38% of the respondents, with an additional 52% indicating conditional interest. Online surveys were the most preferred mode of participation, followed by workshops and board meetings. **Conclusions**: Participatory bilingual approaches are feasible in South Tyrol and highlight priorities that are highly relevant for patient-centered health services. Future initiatives should strengthen the structures for research participation, enhance GP engagement, and link identified priorities to research funding and policy action.

## 1. Introduction

Engaging patients, caregivers, and community groups in determining research priorities is increasingly acknowledged to ensure that health services research addresses genuine needs and enhances patient experiences. Methods such as the James Lind Alliance Priority Setting Partnerships and community-based participatory strategies have proven effective in formulating research questions that differ from those identified solely by researchers, often emphasizing issues such as access to care, service coordination, and self-management [1,2].

Regions characterized by linguistic and cultural diversity present difficulties in prioritizing. Research from places such as Ireland and Singapore, as well as other multilingual areas, indicates that traditional top-down methods might neglect the needs of minority groups. In contrast, well-managed participatory processes can reveal concerns specific to the context and enhance the legitimacy of decisions made [3,4].

Despite these advantages, participatory activities often face obstacles. Disparities in power between researchers and participants can restrict the impact of community contributions, and marginalized groups are frequently underrepresented, even when their inclusion is a stated objective [5]. Additionally, resource limitations and institutional demands, such as inflexible funding structures, hinder ongoing involvement [6].

Research has shown that involving participants in setting priorities can shift research towards topics that focus on patients and enhance stakeholder satisfaction. However, its long-term effects on healthcare quality and outcomes have not been thoroughly documented [7]. It is crucial to evaluate these processes in various settings to create effective strategies for fair participation and determine whether the identified priorities eventually influence policy and service delivery.

Building on this evidence, the present study aimed to identify regional health services research priorities in South Tyrol, a culturally and linguistically diverse region [8,9], through a bilingual participatory survey involving general practitioners (GPs) and patient and social interest organizations (PSIOs).

## 2. Methods

### 2.1. Study Design and Participants

An online cross-sectional survey was conducted to determine the priorities for regional health services research from the perspectives of patients and social interest organizations (PSIOs), as well as general practitioners (GPs) in South Tyrol, Italy. On 25 August 2025, an email invitation was dispatched to all 64 member organizations of the South Tyrolean Umbrella Association for Social and Health Affairs (“Dachverband für Soziales und Gesundheit”), with a reminder sent on 17 September 2025. Five of these email addresses were invalid. Each PSIO was asked to share the invitation with its leaders, staff, and members to increase participation rates. Simultaneously, GPs employed by the South Tyrolean Public Health Service (*n* = 290) received direct invitations. The survey concluded on 25 September 2025.

### 2.2. Questionnaire Development and Content

The questionnaire was developed by the Institute of General Practice and Public Health to address the study objectives, drawing on relevant literature on participatory priority setting and prior health services research conducted in South Tyrol. In addition, the current program activities of the institute—the only dedicated public health research institute in the region—were incorporated to ensure contextual relevance. The questionnaire did not collect personal demographic information, as responses were provided on behalf of organizations rather than individual members. The instrument covered organizational characteristics, respondent roles, priority ratings for a predefined set of regional health research topics, experiences with patient and public involvement in healthcare planning, perceived barriers to healthcare access, suggestions for enhancing patient influence in decision-making, and preferred modes of engagement in priority setting (e.g., online surveys, workshops, focus groups, and patient advisory boards). Open-ended items enabled respondents to raise additional health system issues and to provide free-text recommendations.

The draft questionnaire was prepared in German, translated into Italian by a native speaker, and provided in both languages. An English version is available as Appendix A. It combined multiple-choice, ranking, and open-ended questions to gather information. To ensure clarity and validity, the questionnaire was piloted by a biostatistician, GP, and clinical psychologist before distribution.

### 2.3. Data Collection

The survey was administered using Google Forms (Google LLC, Mountain View, CA, USA; https://forms.google.com). Responses were anonymous, and participation was voluntary and confidential. Completion of the online questionnaire was considered as implied consent. According to local regulations, formal ethics committee approval was not required because no patient-level clinical data were collected for this study.

### 2.4. Statistical Analysis

Analyses were conducted using IBM SPSS Statistics for Windows version 25.0 (IBM Corp., Armonk, NY, USA). All analyses were prespecified as descriptive and exploratory. Categorical variables were summarized as counts and percentages, and ordinal/continuous items as mean (SD) and median (IQR), reporting the valid n for each item (no imputation). Unless stated otherwise, tests were two-sided with α = 0.05; given the study’s exploratory aims, p-values are presented unadjusted and interpreted alongside effect sizes.

The language of free-text responses was recoded to German and Italian; blanks were left as system missing. For multi-response questions (e.g., preferred participation modes; barriers; facilitators), open answers were content coded using a directed approach into binary indicators (1 = theme mentioned; 0 = not mentioned) and summarized as proportions of all respondents. A concise codebook describing all inductively derived categories with translated example quotations is provided in Appendix A.

Because priority ratings were captured on 5-point ordinal scales and group sizes were unequal, GPs versus PSIO respondents were compared using Mann–Whitney U tests for each research topic; medians and IQRs are shown per group. For associations between categorical variables, Pearson’s chi-square tests with Cramer’s V as the effect size (small ≈ 0.10–0.29, moderate ≈ 0.30–0.49, large ≥ 0.50) were used. The association between language and research interest was examined using chi-square, and Kendall’s tau-b was additionally reported as an ordinal measure of association. Missing data were excluded listwise for the variables involved in each specific test.

To explore the relationships between interest in research participation, topic priorities, and inter-topic relationships, Spearman’s rank correlations (ρ) were computed using pairwise complete observations. Correlation magnitudes were interpreted using conventional thresholds (small ≈ 0.10–0.29; moderate ≈ 0.30–0.49; large ≥ 0.50) [10]. Given the large number of inter-item correlations, the false discovery rate (FDR) was controlled using the Benjamini–Hochberg procedure with q = 0.10, a threshold commonly used in exploratory health services research to balance type I and type II errors [11].

Because several PSIOs submitted multiple questionnaires on behalf of the same organization, clustering was examined using the cleaned organization identifier. The mean cluster size among PSIOs was 1.55 (median 1, range 1–31). To quantify potential non-independence, the intraclass correlation coefficient (ICC) for the composite priority score (mean of eleven research-priority items) was estimated with a one-way random-effects model using restricted maximum likelihood. The ICC was calculated as the ratio of between-organization to total variance. A sensitivity analysis was then conducted, restricting the dataset to one response per identifiable organization (first by timestamp) while retaining all GP responses. All analyses were repeated in this reduced dataset to evaluate the robustness of results.

### 2.5. Use of Generative Artificial Intelligence

Generative Artificial Intelligence (AI; ChatGPT-5 model, OpenAI, San Francisco, CA, USA; https://chat.openai.com) was used to assist in structuring and refining the manuscript text, including the formulation of the Introduction and Methods sections. AI was also used to synthesize and cross-reference the existing literature to ensure clarity and contextualization. No generative AI was used for data collection, statistical analysis, or interpretation of the results. All content was reviewed and approved by the authors.

## 3. Results

### 3.1. Survey Participation and Respondent Characteristics

The flow diagram of email invitations, survey submissions, and analyzed questionnaires among PSIOs and GPs is shown as Appendix A. A total of 95 completed questionnaires were analyzed. Of these, 9 respondents (9.5%) were GPs employed in primary care, while the remaining 86 respondents (90.5%) represented PSIOs. Within the PSIO group, participants identified as members or service users (*n* = 38; 40.0%), board members or leaders (*n* = 27; 28.4%), staff (*n* = 15; 15.8%), or other roles (*n* = 6; 6.3%). This distribution confirms that most responses originated from the PSIO sector, with GPs contributing a smaller but distinct subgroup of respondents. Based on invitations distributed, 27 of 59 PSIOs (45.8%) submitted at least one completed questionnaire, whereas 9 of approximately 290 invited GPs (3.1%) participated. As invitations were distributed both to organizational representatives and through organizations to their members, multiple responses per organization were possible. Thus, the distribution reflects individual perspectives rather than a strict one-response-per-organization approach.

Among all responses, 51 respondents (53.7%) provided a valid and identifiable organization name, while 44 (46.3%) either left the field empty or provided information that could not be linked to a specific organization. Of the organizations that could be classified, chronic disease associations were the most frequent (*n* = 28; 54.9%), followed by disability-related groups (*n* = 11; 21.6%), mental health or psychosocial support organizations (*n* = 7; 13.7%), child and family welfare organizations (*n* = 2; 3.9%), and social inclusion or poverty relief groups (*n* = 3; 5.9%).

These categories represent a wide range of patient and community interests. Together, chronic disease and disability organizations accounted for more than three-quarters of all identified PSIO, indicating that the survey successfully engaged the most active segments of the South Tyrolean Federation for Social and Health Affairs.

Among the 59 PSIOs invited, 55 organizations provided at least one response. Most contributed a single questionnaire (43/55), while one umbrella organization contributed 31. The ICC for clustering by organization was 0.50, indicating that half of the variance in mean priority scores was attributable to between-organization differences. Given the mean cluster size of 1.55, the corresponding design effect was 1.28, suggesting a modest reduction in effective sample size. When analyses were restricted to one response per organization (*n* = 58), results for research-interest proportions, GP–PSIO comparisons of priority topics, and inter-item correlations remained directionally unchanged. Only the association for environmentally sustainable and safe prescription of medicines approached but did not reach significance (*p* = 0.07). These findings confirm that clustering did not materially affect the study’s conclusions.

#### 3.1.1. Language of Open-Text Responses and Research Interest

Open-text responses were provided predominantly in German (*n* = 52; 71.2%), while a smaller group responded in Italian (*n* = 21; 28.8%). Twenty-two participants (23.2% of the total sample) left these fields empty and were coded as missing. It should be noted that the questionnaire was available in both German and Italian; the coding here reflects the language used in free-text answers rather than the version of the questionnaire selected. Among German-language responses, most indicated conditional interest (“maybe/more information desired,” 55.8%), followed by definite interest (34.6%) and no interest (9.6%). Italian-language responses showed a different distribution, with a higher proportion expressing definite interest (47.6%), 42.9% reporting conditional interest, and 9.5% indicating no interest. Respondents with missing entries displayed a pattern like the German-language group, with conditional interest being most frequent.

Despite these descriptive differences, a chi-square test of independence did not reveal a statistically significant association between language and research interests (χ^2^ = 1.14, df = 2, *p* = 0.566). The effect size was small (Cramer’s V = 0.125), and ordinal association measures (Kendall’s tau-b = −0.12, *p* = 0.292) confirmed the absence of systematic trends.

#### 3.1.2. Perceived Barriers to Healthcare Access

The open-ended survey question on perceived barriers to healthcare access among survey participants (“What barriers do your members experience in accessing health care?”) was analyzed using the same directed content approach described above. Figure 1 presents the frequency of these content-derived categories as a proportion of all 95 respondents.

Long waiting times clearly dominated the responses, mentioned by more than two-fifths of the participants. A shortage of healthcare personnel was the second most frequent barrier, followed by bureaucratic procedures. Organizational issues, digitalization/IT problems, language barriers, lack of trust in healthcare, financial costs, limited access to specialist care, care transitions, and mobility or transport limitations were reported by progressively smaller proportions of respondents. These findings indicate that, within this mixed group of GPs and PSIOs, barriers are concentrated on delays and human resources, whereas structural, financial, and mobility obstacles are far less prominent.

Respondents detailed how prolonged waits disrupted both routine care and access to specific benefits, sometimes forcing individuals to seek costly private services to avoid waiting. Financial pressures were evident in reports of out-of-pocket payments for dental treatment or essential medications, despite formal fee waivers. Several narratives described the difficulties faced by vulnerable groups, such as people with disabilities, cognitive impairments, or limited language proficiency, who encountered dismissive attitudes or practical obstacles when seeking preventive or specialized services. Structural critiques included the persistence of a hospital-centered care model, fragmented reporting and reimbursement procedures, and electronic health records used mainly as document repositories rather than tools for integrated care.

### 3.2. Research Priority Ratings

Participants evaluated a predetermined list of research topics using a 5-point priority scale.

As illustrated in Table 1, the top priorities were the mental health of children and adolescents, continuity and trust in primary care, and patient-focused palliative care and end-of-life care. These were closely followed by reliable health information, communication, and access to healthcare for disadvantaged groups. Moderate-priority topics included the freedom to choose a trusted general practitioner and its effect on waiting times for specialist appointments and diagnostic procedures, detecting frailty in older adults and appropriate care, and health education in schools. Lower priorities were given to environmentally sustainable and safe prescribing, the family impact of gender differences in stress management, and vaccine acceptance with culturally sensitive information.

Open-text responses highlighted locally relevant themes not covered by the predefined list, including waiting times and access barriers, calls for broader system reform, psychosocial support, attention to vulnerable groups, and preventive health promotion.

Among the 95 valid survey responses, 85 respondents (89.5%) represented PSIO, and 10 respondents (10.5%) were GPs (nine in primary care and one as a PSIO member). Mann–Whitney U tests comparing the priority ratings between GPs and PSIO across all 12 research topics revealed no statistically significant differences (all *p* > 0.16). The mean rank patterns suggest only minor and non-significant tendencies. GPs rated vaccine acceptance and access to disadvantaged groups as slightly higher whereas PSIOs rated frailty and free choice of physician somewhat higher.

Because the GP subgroup was small (*n* = 9), these non-significant results should be interpreted cautiously as reflecting limited statistical power rather than evidence of equivalence or concordance between groups (Figure 2).

### 3.3. Interest in Research Participation

Among the 81 valid responses to the question, “Is your organization interested in participating in research projects (e.g., in formulating research questions, workshops, or sharing experiential knowledge)?”, 38.3% indicated yes, 9.9% responded no, and 51.9% selected maybe/more information desired, while 14 participants (14.7% of the total sample of 95) provided no answer.

When examined by organizational role, definite interest in research participation was most frequent among board or leadership members (46.2%), followed by members or service users (40.7%) and staff (21.4%); GPs showed an intermediate level (33.3%). A chi-square test of independence (χ^2^ = 7.18, df = 12, *p* = 0.846) revealed no statistically significant association between organizational role and research interest, reflecting a small sample size.

#### Correlation Results

Spearman correlation analyses were used to examine the associations between interest in research participation and rated priority topics for health research (Table 2). Given the exploratory purpose of these analyses, correlation findings should be interpreted as descriptive and hypothesis-generating rather than confirmatory. After Benjamini–Hochberg correction at q = 0.10, 52 of 66 pairwise correlations remained significant, indicating that the observed associations were robust to multiple testing control.

Interest in research participation showed small negative correlations with several priority topics, including palliative care (ρ = −0.23, *p* = 0.049), primary care continuity (ρ = −0.26, *p* = 0.022), and stress in families (ρ = −0.25, *p* = 0.034). These results suggest that respondents expressing a higher interest in participating in research tended to assign slightly lower priority ratings to these specific research areas. Other associations with interest in research were weak and insignificant.

Among the priority topics themselves, there was a consistent pattern of positive inter-correlations, indicating that participants who rated one topic as highly important tended to give high ratings to other topics.

The strongest relationships were observed between frailty and palliative care (ρ = 0.66, *p* < 0.001), health education in schools and stress in families (ρ = 0.59, *p* < 0.001), and primary care continuity with appropriate prescribing (ρ = 0.64, *p* < 0.001).

Numerous moderate associations (ρ ≈ 0.30–0.50, all *p* < 0.01) linked most other topic pairs, suggesting a generally coherent pattern of high concern across the research priorities.

### 3.4. Recommendations for Enhancing Patient and Public Involvement

Among the 95 respondents, 15 (15.8%) reported prior involvement of their members or users in the planning or evaluation of health services, while 42 (44.2%) stated that no participation had taken place, and 38 (40.0%) were uncertain. Thirty-five respondents (36.8%) expressed definite interest in participating in research projects, while an additional 47 (49.5%) indicated conditional interest, requesting more detailed information before committing. Only 13 respondents (13.7%) reported no interest in research collaboration. These findings suggest substantial openness within PSIOs and GPs to engage in research in the ways specified in the questionnaire, such as contributing to research questions, participating in workshops or focus groups, or providing feedback on research results, provided that adequate information and support are available.

#### 3.4.1. Feeling Heard in Health Decisions and Research

The open-ended question “What would help your members feel better heard in health decisions and research?” was analyzed using a directed content approach. All narrative responses were screened for recurring themes and subsequently coded into binary variables indicating whether a respondent mentioned a specific theme.

Table 3 presents the frequency of the content-derived categories as the proportion of all 95 respondents in the study. Quantitative coding showed that clearer communication between decision-makers and the public was the most frequently mentioned facilitator (12.6%), followed by more time for participation (10.5%) and active inclusion of patients or community representatives (7.4%). Other suggestions included representation in formal governance bodies and greater recognition of general practitioners. Less commonly reported were calls for better education and information to support engagement, greater empathy from institutions, improved accessibility or barrier-free opportunities, reduced bureaucratic barriers, and increased trust in health authorities or researchers.

Requests for “communication on equal terms” and the use of clear, simple language, especially for people with intellectual disabilities or limited language proficiency, were common. As one participant put it, “General practitioners should have more time to listen to patients.’ Another emphasized that patients “should be perceived as persons, not as numbers.” Respondents highlighted the need for more time with physicians and other health professionals to allow for genuine listening and dialog beyond routine consultations.

Several comments called for formal mechanisms that would give patients and caregivers a recognized voice in decision-making. Suggestions included the creation or revival of decision-making bodies that include “experts by experience,” systematic integration of patient organizations into planning processes, and designated contact persons within the health authority to address concerns. Some respondents asked for greater recognition of general practitioners within the regional health service and for the simplification of bureaucratic procedures, such as easier booking systems or more transparent administrative pathways.

Participants advocated for improved health education and public campaigns, such as organ donation awareness and better information on chronic diseases, to empower citizens to participate effectively. One long and detailed statement called for “systematic integration of patient organizations; transparent communication and co-design; participation in clinical research through patient advisory boards; and structured opportunities for feedback, such as patient portals or satisfaction surveys, that are taken seriously and used to achieve concrete improvements.”

#### 3.4.2. Preferred Modes of Participation in Setting Research Priorities

The survey asked respondents how they would prefer to participate in defining research priorities (“How would you prefer to participate in setting research priorities?). Select up to two options”). All answers were content coded into binary variables to capture the presence of each predefined option, allowing multiple selections per respondent.

Figure 3 displays the proportion of all 95 respondents who selected each participation mode. Online surveys were the most frequently chosen format, followed by participation in in-person or online workshops and involvement in patient advisory boards. Roughly one quarter indicated interest in focus groups and one fifth in providing feedback on the research results. A small minority (5.3%) provided free-text suggestions outside the predefined options, mainly calling for direct personal dialog with researchers or hybrid formats that combine elements of surveys and workshops.

## 4. Discussion

This priority-setting survey demonstrated stakeholder engagement and clarity of the research agenda. Although most respondents represented PSIOs, leadership positions were well represented, providing perspectives from individuals pivotal to patient involvement and citizen science. Across this diverse group, long waiting times and shortages of healthcare personnel emerged as the most pressing barriers to healthcare access, while the highest research priorities focused on the mental health of children and adolescents, continuity and trust in primary care, and patient-centered palliative and end-of-life care. Interest in active participation in research was substantial, with nearly 40% expressing definite willingness and more than half requesting additional information, indicating potential partners for collaborative research and health-system development. These findings highlight the need for collective awareness of system bottlenecks, convergent research priorities, and readiness among PSIOs to engage in future participatory research efforts.

### 4.1. Respondent Characteristics

The linguistic distribution of the responses reflects South Tyrol’s demographics, where approximately 68% of the population is German-speaking and 27% Italian-speaking [12]. In our sample, 71.2% of the answers were in German and 28.8% in Italian. This indicates that the survey captured the region’s two majority language groups while potentially underrepresenting linguistic minorities, possibly including participants with missing free-text responses.

Interest in participatory research did not differ between German- and Italian-speaking respondents, showing linguistic background did not affect research collaboration willingness. Evidence suggests language alone does not influence participation when engagement methods are inclusive. Successful engagement across linguistic groups depends more on quality facilitation, trust-building, and accessible materials than specific language [13,14,15,16].

Common barriers to healthcare access—long waiting times, staff shortages, and bureaucratic procedures—indicate system-level challenges affecting the population. Less frequent obstacles like language barriers, trust issues, and mobility limitations highlight burdens faced by disadvantaged groups. Long waiting times, staff shortages, and bureaucratic procedures represent system-level obstacles that disproportionately affect those with fewer resources [17,18]. Such barriers discourage access to care by contributing to perceptions of inaccessibility [18]. These findings demonstrate why perceived barriers matter: they shape participation conditions and require addressing. Language barriers, trust issues, and mobility limitations particularly affect disadvantaged groups like migrants and people with disabilities, who risk underrepresentation without targeted measures [19,20]. Addressing these barriers through trust-building and accessible formats is essential for equitable inclusion [18,20].

### 4.2. Interpretation of Research Priority Patterns

The identified prioritization patterns highlight challenges for healthcare systems, with youth mental health, continuity of primary care, and palliative care receiving the highest rankings. The urgency of youth mental health reflects a global crisis exacerbated by COVID-19, with increasing psychological distress and unmet needs among adolescents and young adults [21,22,23]. Evidence shows that delayed detection and care lead to worse outcomes, underscoring the need for integrated, accessible, and youth-friendly primary care models [24,25]. Continuity in primary care was prioritized, consistent with findings that sustained provider relationships reduce hospitalizations, improve care transitions, and strengthen patient outcomes [26,27]. The high priority for palliative care reflects the growing recognition of patient-centered approaches in serious illness management, particularly in improving the quality of life and aligning care with patient values, although mental health support remains insufficient [28,29].

Reliable health information and equitable access for disadvantaged groups have emerged as priorities, highlighting their role in reducing inequities and responding to systemic challenges. International evidence shows that socioeconomically disadvantaged, minority, rural, and older populations face barriers, including limited digital access and skills [30,31], low health literacy [32,33], and structural obstacles such as language barriers, discrimination, and limited trust in providers [34,35]. These findings mirror respondents’ concerns and emphasize the need for culturally tailored interventions that engage affected groups in health information design and delivery [32,36].

Moderate priorities, such as GP choice, frailty detection, and school health education, were rated secondary to urgent issues, whereas sustainable prescribing, gender stress differences, and vaccine acceptance received lower priority. This suggests that respondents viewed the immediate challenges of mental health, care continuity, and access as more pressing than structural or long-term issues.

The absence of significant differences between GPs and PSIOs indicates a consensus across stakeholder groups rather than profession-specific agendas. Similar findings have been reported in structured priority-setting processes, where clinicians and patient stakeholders emphasize quality of care, access, and care models [37,38,39]. Evidence from cancer and general practice research shows convergence, with prevention, survivorship, and care coordination being rated highly across groups [40]. This alignment strengthens the legitimacy of the present results as a foundation for future participatory research. However, given the very small GP stratum, the absence of statistically significant differences should not be interpreted as confirming alignment between GPs and PSIOs but rather because of limited power to detect modest effects. When priorities are shared, the resulting agendas are more inclusive, less vulnerable to claims of narrow professional bias, and more likely to attract support, funding, and implementation [2,37,41]. Broad agreement across stakeholders thus enhances the credibility of identified priorities and underscores the value of participatory approaches in ensuring that health research addresses community-relevant concerns.

### 4.3. Interpretation of Interest in Research Participation

Interest in future research participation was moderate, with approximately 38% of respondents expressing willingness, 10% declining, and just over half reporting uncertainty. This distribution mirrors findings from other contexts, where many potential participants remain undecided until they receive clearer information about the study’s aims, procedures, and benefits [42,43,44]. Uncertainty often reflects not a lack of interest but the need for greater trust, reassurance about ethical safeguards, and practical support to overcome perceived barriers to participation [45,46]. Evidence shows that transparent communication, trust-building, and reducing the logistical burden of participation can help translate moderate levels of willingness into active engagement [42,47,48].

PSIO board and leadership members showed more interest in future research participation than staff, with GPs between, though differences were not statistically significant. This aligns with studies showing professional role alone does not predict research engagement [49,50]. GPs have been described as difficult to recruit, with willingness varying by age, gender, and practice context [51]. Education, prior research experience, and organizational support are stronger determinants of participation, highlighting the importance of fostering a supportive research culture across professional roles in nursing.

Correlation analyses revealed minor negative associations between participation willingness and priority ratings for palliative care, continuity of care, and family stress, while strong positive correlations existed among other topics. Willingness to engage in research was largely independent of topic preferences, though those rating sensitive areas as priorities appeared less inclined to participate. This aligns with evidence that emotionally taxing research topics may reduce participation willingness [43,44,51].

The correlations between topics like frailty and palliative care, school health education and family stress, and primary care continuity and prescribing, indicate their interconnected nature. This clustering reflects a systems perspective, emphasizing integrated health approaches, consistent with priority-setting studies showing related theme groupings [52,53]. The results support that these research areas were perceived as jointly important, underscoring the coherence of the identified agenda.

### 4.4. Preferred Modes of Research Participation

In this study, online surveys have emerged as the preferred mode of participation, reflecting convenience, flexibility, and privacy. This finding aligns with evidence that mode preference predicts actual participation in web-based studies [54,55,56]. Workshops and patient advisory boards were frequently mentioned, suggesting participants value interactive and collaborative formats, despite greater time commitment and institutional support [52,57]. Focus groups and feedback on research results were less prioritized, indicating dialogic approaches appeal to a smaller subgroup [58,59]. Finally, only a minority preferred free-text or hybrid formats, showing that while flexibility is welcomed, most participants prefer structured modes [52,60].

### 4.5. Recommendations for Enhancing Research Participation

While few respondents had prior structured patient involvement experience, most were willing to engage in future research activities, suggesting the potential for participatory approaches with proper support. Evidence shows that effective engagement relies more on creating conditions that empower contributors and establish inclusive partnerships than on baseline familiarity [52,61,62,63,64].

Respondents emphasized clear communication, dialog time, and inclusion strategies, aligning with best practices for patient involvement in health research. Transparent communication about involvement in activities builds trust [52,64,65]. Regular feedback and demonstration of how input shapes decisions strengthen engagement.

Sufficient time and resources enable meaningful engagement, as tokenistic processes risk limiting impact [52,64]. Diversity requires flexible meeting formats and support for patients, PSIOs, and GPs [66,67]. Recognizing each group’s expertise, particularly GPs’ practical knowledge, bridges professional and community perspectives [52,67]. Shared decision-making approaches that respect all voices lead to greater trust and research impact [61,65].

In line with our results, where only 15.8% reported prior involvement but most expressed definite or conditional willingness, education and training can build confidence and capacity, while empathetic engagement and accessible processes lower barriers to entry [68,69,70]. Conversely, concerns about administrative burdens and institutional transparency reflect well-documented obstacles that deter participation, underscoring the need for structural solutions and feedback mechanisms to translate willingness into active involvement [68,69].

### 4.6. Strengths and Limitations

This study had several strengths. This study represents the first structured, bilingual, participatory survey on health services research priorities in South Tyrol, a region marked by linguistic and cultural diversity. The inclusion of both German- and Italian-speaking organizations ensured coverage of the region’s two main language groups, and the involvement of PSIOs provided perspectives beyond those of professionals. With 64 PSIOs active in a region of approximately 530,000 inhabitants, their engagement provides a broad basis for exploring community-informed research priorities. The bilingual design, piloted by experts with clinical and methodological backgrounds, and the combination of structured items with open-ended questions likely enhanced contextual relevance and validity.

This study has several limitations must be acknowledged. Most importantly, the response rate among GPs was alarmingly low, with only nine of 290 invited GPs responding. This likely reflects a combination of factors, including the reliance on a single email invitation and one reminder, the risk that messages were overlooked, and the well-documented challenges of engaging GPs in research, such as limited time and the perceived low relevance of academic studies to daily practice [51,53]. Consequently, the GP perspective is underrepresented in the findings, and observed similarities between groups should not be interpreted as statistical or population-level agreement.

The predefined list of research topics was investigator-seeded, informed by prior literature and program activities, which may have introduced framing bias. Open-ended items partially mitigated this by allowing respondents to propose additional issues, such as prevention, digital health, and social support needs, although other locally salient priorities may still have been missed. The questionnaire language was not systematically recorded; therefore, analyses used the language of free-text responses as a proxy, which may have caused misclassification. Language-based comparisons should thus be interpreted with caution. Furthermore, individual demographic information such as age or gender was not collected, as responses were provided on behalf of organizations rather than individual members. Consequently, it was not possible to assess whether demographic characteristics influenced the prioritization of research topics. This limits the interpretation of potential subgroup differences in perceived importance.

Given the exploratory nature of this study, the Benjamini–Hochberg FDR correction (q = 0.10) was applied as a sensitivity approach. While several inter-topic associations remained significant after adjustment, marginal effects should be viewed as indicative rather than confirmatory [11].

Finally, as a cross-sectional survey, the study provides a snapshot of current perceptions and research priorities. Longitudinal or follow-up studies would be needed to determine how these priorities evolve over time or whether they ultimately influence health policy or practice.

## 5. Conclusions

This study provides the first systematic bilingual assessment of health service research priorities in South Tyrol, engaging both GPs and PSIOs in a participatory process. Despite the very low GP response rate, the participation of community organizations highlights the feasibility and value of including diverse stakeholders in regional priority setting.

The findings emphasize the centrality of child and adolescent mental health, continuity and trust in primary care, and palliative and end-of-life care as top priorities, alongside enduring concerns about equitable access and reliable health information. While participatory approaches revealed broad agreement, the study also uncovered practical barriers to engagement, including limited prior involvement and a preference for low-burden participation modes, such as online surveys.

These results demonstrate that participatory approaches can generate priorities that are context-specific, patient-centered, and relevant to health policy and service development in a culturally diverse region. Future efforts should build on this foundation by improving the recruitment of healthcare professionals, strengthening structural mechanisms for patient and public involvement, and ensuring that identified priorities are systematically linked to research funding and policy decisions.

## Figures and Tables

**Figure 1 healthcare-13-02797-f001:**
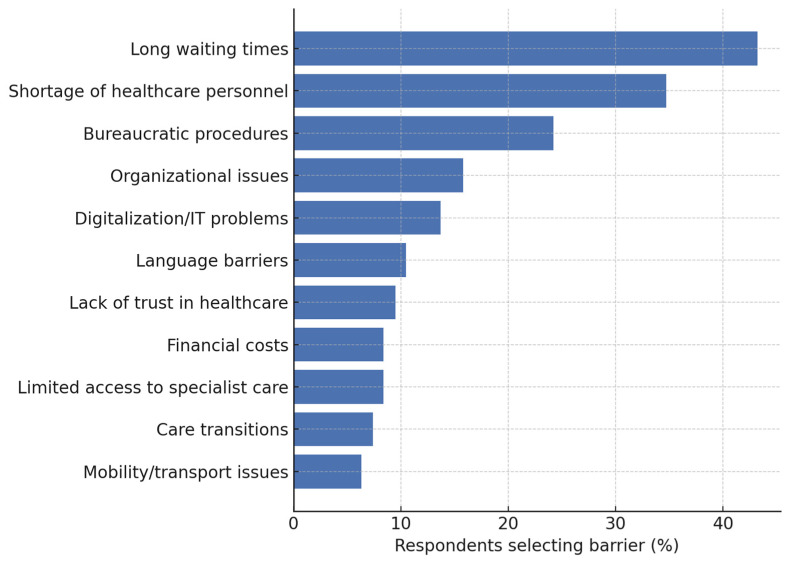
Barriers to healthcare access reported by the survey respondents (*n* = 95); no missing values were observed. Categories were derived inductively from open-ended responses via open coding and subsequently binary-coded for frequency analysis. Horizontal bar chart showing the percentage of respondents who selected each predefined barrier.

**Figure 2 healthcare-13-02797-f002:**
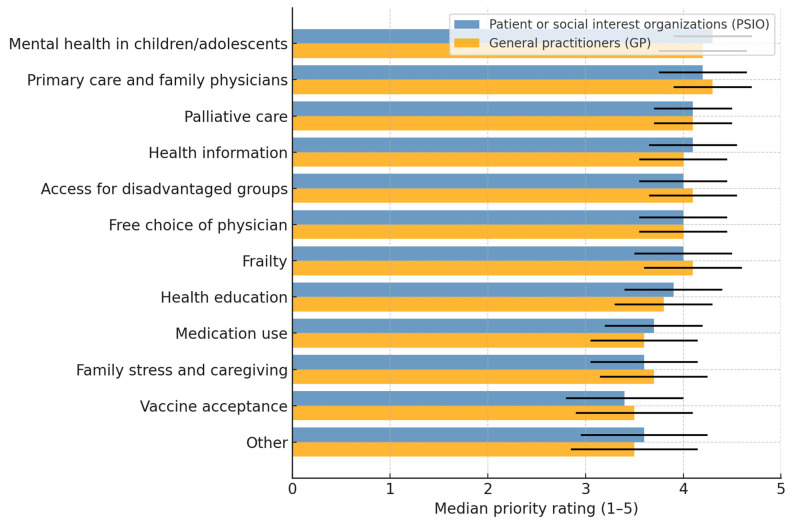
Median priority ratings for investigator-seeded research topics among PSIOs and GPs. Research topics were predefined based on literature and regional program activities. Median ratings (1 = lowest to 5 = highest priority) with interquartile ranges (black lines) are shown for each research topic. Mann–Whitney U tests, *p* > 0.05.

**Figure 3 healthcare-13-02797-f003:**
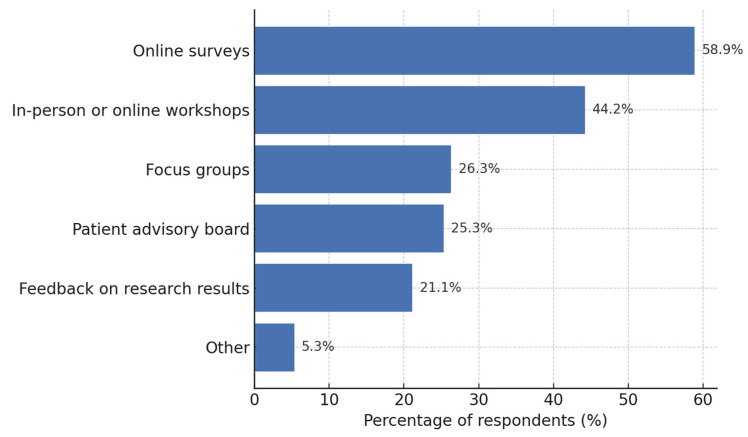
Preferred modes of participation in setting research priorities (*n* = 95). Participation formats were predefined Free-text responses suggesting additional or hybrid formats were coded inductively and summarized under “Other”. Horizontal bar chart showing the percentage of respondents who selected each option.

**Table 1 healthcare-13-02797-t001:** Descriptive ranking of investigator-seeded research priority topics.

Priority Topic ^1^	Rank by Mean	*n* (Valid) ^2^	Mean (SD)	Median (IQR)
Mental health of children and adolescents	1	87	4.29 (1.01)	5.0 (4–5)
Continuity and trust in primary care	2	88	4.15 (1.07)	5.0 (4–5)
Patient-oriented palliative and end-of-life care	3	85	4.11 (1.02)	4.0 (4–5)
Trustworthy health information and communication	4	89	4.09 (1.16)	5.0 (4–5)
Access to healthcare for disadvantaged groups	5	84	4.02 (1.13)	4.0 (4–5)
Impact of free choice of a trusted GP on waiting times for specialist appointments and diagnostic procedures	6	87	4.00 (1.19)	5.0 (4–5)
Detection of frailty in older age and appropriate care	7	89	3.96 (1.17)	4.0 (3–5)
Health education in schools	8	89	3.91 (1.14)	4.0 (3–5)
Environmentally sustainable and safe prescription of medicines	9	86	3.69 (1.20)	4.0 (3–5)
Family consequences of gender differences in coping with stress/psychological burden)	10	85	3.58 (1.23)	4.0 (3–4)
Vaccine acceptance and culturally sensitive information	11	87	3.44 (1.37)	4.0 (2–4)
Other topics specified in free text ^3^	—	19	3.63 (1.80)	5.0 (2–5)

^1^ Research topics were predefined based on literature and regional program activities. ^2^ Valid *n* varies due to missing values. ^3^ Open-ended responses were optional and thus less comparable across the participants. Abbreviations: SD, standard deviation; IQR, interquartile range.

**Table 2 healthcare-13-02797-t002:** Spearman correlations between research participation interest and priority topics (and among the priority topics).

Variables	*n* (Pairs)	ρ (Spearman) *	*p*-Value	Effect Size ^†^
Interest in research participation × Palliative care priority	73	−0.23	0.049	Small
Interest in research participation × Primary care continuity	75	−0.26	0.022	Small
Interest in research participation × Stress in families	73	−0.25	0.034	Small
Frailty × Palliative care	84	0.66	<0.001	Large
Health education in schools × Stress in families	82	0.59	<0.001	Large
Primary care continuity × Appropriate prescribing	85	0.64	<0.001	Large
Frailty × Primary care continuity	84	0.46	<0.001	Moderate
Frailty × Appropriate prescribing	84	0.44	<0.001	Moderate
Palliative care × Primary care continuity	83	0.48	<0.001	Moderate
Palliative care × Appropriate prescribing	82	0.45	<0.001	Moderate
Health education in schools × Mental health of children	85	0.42	<0.001	Moderate
Access for disadvantaged groups × Primary care continuity	83	0.39	0.002	Moderate
Access for disadvantaged groups × Appropriate prescribing	82	0.37	0.003	Moderate

* Negative values indicate that a higher research participation interest is associated with lower priority ratings for the respective topic. ^†^ Effect size categories follow guidelines for correlation coefficients: small ≈ 0.10–0.29, moderate ≈ 0.30–0.49, large ≥ 0.50 [10].

**Table 3 healthcare-13-02797-t003:** Measures suggested to help members feel better heard in health decisions and research (*n* = 95).

Improvement Category ^1^	*n*	% of Respondents ^2^
Clearer communication	12	12.6%
More time for participation	10	10.5%
Active inclusion of patients/community	7	7.4%
Representation in formal governance bodies	5	5.3%
Greater recognition of GPs	5	5.3%
Better education and information	4	4.2%
Greater empathy from institutions	3	3.2%
Improved accessibility/barrier-free opportunities	3	3.2%
Reduction in bureaucratic barriers	2	2.1%
Increased trust in health authorities/researchers	2	2.1%

^1^ Categories were derived inductively from open-ended responses using open coding. Each theme was subsequently coded as a binary variable (1 = mentioned, 0 = not mentioned) for descriptive frequency analysis. ^2^ Percentages are calculated based on valid responses for each item (*n* = 95); no missing values were observed.

## Data Availability

The original contributions presented in this study are included in the article/Appendix A. Further inquiries can be directed to the corresponding author.

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
