# Peer review of "Patients and Communities Shape Regional Health Research Priorities: A Participatory Study from South Tyrol, Italy"

_healthcare, 2025, doi:10.3390/healthcare13212797_

Round 1
Reviewer 1 Report
Comments and Suggestions for Authors
Dear Authors,
Please, see below few comments you might consider or clarify:
1) Participation, counting, and study flow
Reconcile GP counts (e.g., “9 of 95” vs “10 participated”) and report precise denominators for all percentages.
Report response rates by stratum (e.g., PSIO vs GP) using clear numerators/denominators.
Use past tense for all completed data-collection activities and provide a CONSORT-style flow (invitations sent, bounces, opens/clicks if available, starts, completes) separately for PSIO and GPs.
Multiple responses per organization introduce clustering and threaten independence; quantify potential clustering and run a sensitivity analysis restricted to one response per identifiable organization, reporting how estimates change.
Required actions: Reconcile counts; add flow diagram and stratum-specific response rates; conduct and summarize a one-response-per-organization sensitivity analysis.
2) Inference and multiple testing
Non-significant results in very small strata (e.g., 9–10 GPs) primarily reflect low power and should not be interpreted as evidence of concordance or equivalence.
Multiple hypothesis tests were performed; speculative statements about FDR control should be replaced by an actual procedure applied to the full p-value family.
Required actions:
(a) Relocate inferential results to an appendix or label them explicitly as exploratory.
(b) Apply and report a multiple-testing correction (e.g., Benjamini–Hochberg) or avoid binary “significant/non-significant” framing.
(c) Replace claims of “agreement” with descriptive effect sizes and uncertainty; do not infer concordance from non-significance.
3) Framing and measurement bias (priorities and language)
A predefined topic list (seeded from literature/program activities) introduces framing bias and may omit locally salient priorities; claims about “community-driven” agendas should be tempered.
Language analyses used a proxy (language of free-text) rather than the actual questionnaire language, creating misclassification risk and weakening inferences.
Required actions:
(a) Characterize the list as investigator-seeded; highlight the role of open-ended items and report any new topics that emerged.
(b) Analyze by questionnaire language (if captured) or remove language-based comparisons and state the limitation.
4) Denominators and coding for multi-response items
For multi-response items (e.g., barriers/facilitators), binary coding summarized as a proportion of all respondents can understate frequencies when item nonresponse is present; denominators must be explicit and consistent.
Figure captions and Methods should align (predefined vs open-coded lists); current phrasing is inconsistent.
Required actions: Report item-level denominators alongside percentages; clarify whether categories were predefined or derived via open coding in each figure/table; include a concise codebook (with examples) in the Supplement.
5) Strength of claims and causal language
Statements about “broad agreement across stakeholders” are not warranted with a small, non-probability clinician sample.
Causal or demonstrative wording (“demonstrate feasibility/policy relevance”) should be softened.
Required actions: Temper to descriptive language (e.g., “patterns appear broadly similar among respondents”); use “suggest” or “are consistent with” instead of “demonstrate,” and avoid implying representativeness or policy impact beyond the design’s support.
Best wishes
Reviewer 2 Report
Comments and Suggestions for Authors
Thank you for the opportunity to review this manuscript. Please find my comments.
Title
“Perspectives of Patient and Social Organizations in Regional Health Services Research: Priority-Setting Results from SouthTyrol, Italy.” Please, the title could be more attractive.
Objective
Lines 27-30: “ This study aimed to identify health services research priorities in South Tyrol through a bilingual participatory survey involving general practitioners (GPs) and patient and social interest organizations (PSIOs).” / Lines 76-78: “Building on this evidence, the present study investigates how a structured bilingual participatory survey can inform health services research priorities in South Tyrol a culturally and linguistically diverse region [8,9].” Please, align the objectives.
Material and methods
Line 93: “The survey will conclude on September 25, 2025.” OR “The survey was concluded on September 25, 2025.” ? Please, correct the information.
Results
Please, describe demographic characteristics of the participants (age, gender…).
Discussion
Lines 403-415: “Reliable health information and equitable access for disadvantaged groups have emerged as priorities, highlighting their role in reducing inequities and responding to systemic challenges. International evidence shows that socioeconomically disadvantaged, minority, rural, and older populations face barriers, including limited digital access and skills [29,30], low health literacy [31,32], and structural obstacles such as language barriers, discrimination, and limited trust in providers [33,34]. These findings mirror respondents' concerns and emphasize the need for culturally tailored interventions that engage affected groups in health information design and delivery [31,35]. Moderate priorities, such as GP choice, frailty detection, and school health education, were rated secondary to urgent issues, whereas sustainable prescribing, gender stress differences, and vaccine acceptance received lower priority. This suggests that respondents viewed the immediate challenges of mental health, care continuity, and access as more pressing than structural or long-term issues.” Could the age and gender of the participants have interfered with the priorities? Please, discuss this in the text. If it is not possible, this is a limitation of the study.
Round 2
Reviewer 1 Report
Comments and Suggestions for Authors
Thank you for addressing the comments